# Characteristics of Patients Managed without Positive Pressure Ventilation While on Extracorporeal Membrane Oxygenation for Acute Respiratory Distress Syndrome

**DOI:** 10.3390/jcm10020251

**Published:** 2021-01-12

**Authors:** Nicholas M Levin, Anna L Ciullo, Sean Overton, Nathan Mitchell, Chloe R Skidmore, Joseph E Tonna

**Affiliations:** 1Division of Emergency Medicine, Department of Surgery, University of Utah Health, Salt Lake City, UT 84132, USA; Nicholas.Levin@hsc.utah.edu (N.M.L.); Anna.Ciullo@hsc.utah.edu (A.L.C.); u0837115@utah.edu (N.M.); 2Division of Cardiothoracic Surgery, Department of Surgery, University of Utah Health, Salt Lake City, UT 84132, USA; Chloe.skidmore@hsc.utah.edu; 3Division of Critical Care, Department of Anesthesiology, University of Utah Health, Salt Lake City, UT 84132, USA; sean.overton@hsc.utah.edu

**Keywords:** extubation, ventilation, spontaneously breathing, ARDS, respiratory failure, ECMO

## Abstract

Background: Extracorporeal membrane oxygenation (ECMO) has expanding indications for cardiopulmonary resuscitation including severe acute respiratory distress syndrome (ARDS). Despite the adjunct of ECMO for patients with severe ARDS, they often have prolonged mechanical ventilation and are subject to many of its inherent complications. Here, we describe patients who were cannulated for venovenous (VV) ECMO and were taken off positive pressure ventilation. Methods: This is a primary analysis of patients admitted at a tertiary medical center between the dates of August 2014 to January 2020 who were cannulated to ECMO for refractory respiratory failure. We included all patients ≥18 years old. Patients who were extubated or had a tracheostomy and taken off positive pressure while on ECMO were classified as “off positive pressure ventilation (PPV)” and were compared to patients who remained “on PPV” while on ECMO. Primary outcome was survival to hospital discharge. Secondary outcomes were ventilator free days at 30 days and 60 days after ECMO cannulation, time from cannulation to date of first out-of-bed (OOB), and hospital length of stay (LOS). Patient characteristics were derived from routine clinical information in the electronic health record (EHR). Categorical characteristics were compared using chi-square test or Fisher exact test. Continuous characteristics were compared using independent samples t-test or Wilcoxon–Mann–Whitney test. *p*-values were reported from all analysis. Results: Sixty-five patients were included in this retrospective analysis. Forty-eight were managed on ECMO with PPV and 17 patients were removed from PPV. Patients removed from PPV had significantly higher lung injury scores prior to cannulation (2.5 ± 0.6 vs. 1.04 *±* 0.3; *p =* 0.031) and non-significantly longer duration of ventilation prior to ECMO (6.1 days *±* 2.1 vs. 5.0 days *±* 01.1; *p =* 0.634). One hundred percent (100%) of patients removed from PPV survived to hospital discharge compared to 45% who received PPV throughout their duration of ECMO management (*p <* 0.001). The mean ventilator free days at day 60 was 15 with PPV and 36 without PPV (*p =* 0.003). The average duration from cannulation to mobilization (i.e., out-of-bed) was 18 days with PPV and 7 days without PPV (*p =* 0.015). Conclusions: Patients taken off PPV while on ECMO had a very high likelihood of survival to discharge and were mobilized in half as many days. While this likely reflects patient selection, the benefit of early mobilization is well documented and the approach of extubating while on ECMO warrants further investigation.

## 1. Introduction

Unique to patients managed with extracorporeal membrane oxygenation (ECMO) is the ability to provide oxygenation and ventilation independent of the mechanical ventilatory circuit. As such, the additional contribution of gas exchange through the ECMO circuit may reduce the dependence on mechanical ventilation. For patients with severe acute respiratory distress syndrome (ARDS), preventing additional ventilator induced lung injury (VILI) becomes paramount to their management. Previous studies have demonstrated a mortality benefit with a lung protective ventilatory strategy in patients with ARDS, including low tidal volumes and plateau pressure limits [1]. With the use of ECMO in this patient population, the reduction or absence of positive pressure ventilation (PPV) may further improve outcomes.

As the application of ECMO continues to expand [2], including various cardiopulmonary pathologies [3,4,5], a growing interest has emerged in using an awake, non-intubated, spontaneously breathing therapeutic strategy [6,7,8]. Along with limiting ventilator-associated complications in this management approach, extubation also facilitates a reduction in sedation, thereby promoting wakefulness and increasing the opportunity for mobilization [9]. Avoidance of sedating agents in the awake patient also decreases the incidence of delirium, a condition associated with prolonged hospital and intensive care unit (ICU) length of stay as well as increased mortality [10]. While there is literature to suggest the safety and feasibility of extubation while on venovenous (VV) ECMO, studies are limited to ≤12 patients [6,11,12], and/or are among patients with chronic respiratory failure as a bridge to transplant [13,14]. In this study, we look to describe the characteristics of a large group of patients with ARDS managed on ECMO who achieved breathing without the support of positive pressure ventilation.

## 2. Experimental Section

The University of Utah Hospital is a tertiary referral medical center for Utah, Wyoming, Idaho and Nevada. The medical center’s Cardiovascular Intensive Care Unit (CVICU) serves an integral part of the region’s peri-cardiac and mechanical circulatory support intervention site. The CVICU serves as the specific unit responsible in the care of all ECMO cases at this institution. Patients who were admitted to this service and placed on ECMO during their admission were the subjects of interest.

### 2.1. Data Source and Study Population

This secondary analysis was approved by the Institutional Review Board at the University of Utah under #00101562. Data were obtained from a manually maintained research database of patients on ECMO support at a single tertiary academic medical center. Patients are entered into the database by a trained research coordinator blinded to the goals of this analysis at the time of data extraction. The database was previously validated as sufficiently accurate for research and published [15,16]. Patients were identified if they were admitted to the CVICU from August 2014 to January 2020 and were cannulated for ECMO during the index visit. We included all patients ≥18 years of age at the time of admission to the CVICU. All data included was collected as part of routine clinical care.

### 2.2. Study Variables and Outcomes

Our primary outcome is survival to hospital discharge. Secondary outcomes include ventilator free days at 30 days (VFD30) and 60 days (VFD60) after ECMO cannulation, time from cannulation to date of first out-of-bed (OOB), and hospital length of stay (LOS). The first ventilator free day was defined as the first day not requiring positive pressure ventilation, without subsequent return to positive pressure ventilation. The “Off PPV” cohort was defined as patients extubated on ECMO or patients with tracheostomy were removed from positive pressure ventilation while on ECMO versus (vs.) the “PPV” cohort, which was defined as patients who were extubated after ECMO decannulation or tracheostomy patients who were taken off positive pressure only after ECMO decannulation.

Patient characteristics and calculated outcomes included age, sex, baseline medical problems, duration of mechanical ventilation prior to ECMO initiation, lung injury score (Murray Score) [Range 1 to 4 (1: less severe, 4: most severe)] prior to ECMO initiation, ECMO duration, duration of mechanical ventilation, VFD30 and VFD60 starting from date of intubation, survival to hospital discharge, length of stay (LOS) among survivors to hospital discharge, ventilator settings daily for first 3 days on ECMO, Riker Sedation Agitation Scale (SAS) [17] (1 to 7 scale (1 = unarousable, 2 = very sedated, responsive only to physical stimuli, 3 = sedated, responsive to verbal stimuli/gentle touch, 4 = calm and cooperative, 5 = agitated, 6 = very agitated, 7 = dangerously agitated), average time to sedation of ≥11T, net (intake–output) daily fluid balance, time from ECMO cannulation to OOB, time from ECMO cannulation to extubation, time from extubation to hospital discharge and hospital discharge location [home, long term acute care (LTAC), skilled nursing facility (SNF), rehabilitation, Veteran Affairs Hospital (VA), another non-VA Hospital, other].

### 2.3. Statistical Analysis

Our primary goal was to characterize the patient characteristics and outcomes of patients who were either extubated or had a tracheostomy and no longer required positive pressure compared to patients that did not reach those milestones until after ECMO decannulation. We first describe baseline clinical and demographic characteristics of all patients as well as the two cohorts. Descriptive statistics, including counts and percent for binary variables, means [standard error; SE] for continuous variables were used to assess these characteristics. Categorical characteristics were compared using chi-square test or Fisher exact test. Continuous characteristics were compared using independent samples t-test or Wilcoxon–Mann–Whitney test. *p*-values were reported from all analysis. Statistical analyses were conducted in STATA 15.1 (College Park, TX, USA), significance was assessed at the 0.05 level, and all tests were two-tailed.

## 3. Results

A total of 65 patients were included in this retrospective analysis undergoing VV ECMO. The patients baseline characteristics are represented in Table 1 with an average age of 44 and predominantly male (70%). Forty-eight were managed on ECMO with positive pressure ventilation and 17 patients were removed from positive pressure ventilation.

In those managed without positive pressure ventilation, 82% (14) were male vs. 17% (3) who were female (*p =* 0.222). Patients removed from PPV had significantly higher lung injury (Murray Scores) prior to cannulation (2.5 *±* 0.6 vs. 1.04 *±* 0.3; *p =* 0.031), non-significantly longer duration of ventilation prior to ECMO (6.1 *±* 2.1 days vs. 5.0 *±* 1.1 days; *p =* 0.634) as described in Table 2.

Of patients removed from PPV, 17 of 17 (100%) survived to hospital discharge compared to 22 of 48 patients (45%, *p <* 0.001) who received PPV throughout their duration of ECMO management (Table 3). The mean ventilator free days at 30 days (VFD30) was 5.9 *±* 1.6 days in the PPV groups vs. 12.5 *±* 3.2 days in the non-PPV group (*p =* 0.049). VFD60 was 15.5 *±* 3.7 days among patients remaining on PPV vs. 36.9 *±* 5.3 days among patients off PPV (*p =* 0.003). Patients removed from PPV had a positive fluid balance by day three of only 176 mL compared to the patients who remained on PPV with 761 mL (Figure 1).

The average duration from cannulation to mobilization (i.e., out-of-bed) was 18 days with PPV and 7.1 days without PPV (*p =* 0.015, Figure 2). There was no statistical difference between groups based on the Riker Sedation Agitation Score (PPV 2.8 vs. non-PPV 3.6; *p =* 0.084) or in the mean duration until GCS of ≥11T (8.1 days with PPV vs. 9.7 days without PPV; *p =* 0.553). The mean duration of hospital LOS (among survivors) was 41 days in patients receiving PPV and 31 in those removed from PPV (*p =* 0.222), with a majority of patients (38%) discharged to home, regardless of PPV status.

## 4. Discussion

Management of the awake, non-intubated, spontaneously breathing patient while on ECMO provides the benefit of additive gas exchange with the potential of reduced ventilatory-associated complications and early mobilization. The success of this treatment strategy has been described in patients on ECMO as a bridge to lung transplant [9,13,14], but studies among patients with primarily ARDS and VV ECMO are limited to ≤12 patients [6,11,12]. In this manuscript, we report the outcomes of patients managed without PPV while on VV ECMO.

The decision to remove positive pressure ventilation while on ECMO is influenced by several factors which are not measured in our study. Therefore, it is not possible for us to draw conclusions of relative benefit or superiority. We do, however, describe the observed clinical characteristics of patients in whom the decision to remove positive pressure ventilation was made in order to distinguish differences and subsequent outcomes in patients who remained on positive pressure ventilation throughout their ECMO duration. We observed that 100% of the patients who were removed from positive pressure ventilation while on ECMO survived to hospital discharge, compared to 45% of patients who remained on positive pressure ventilation. Additionally, the number of ventilator free days at both the 30- and 60-day measurement were at least doubled in the non-PPV group when compared to the patients that remained on PPV. To our knowledge, this represents one of the larger observational cohorts describing the management of awake, extubated VV ECMO patients.

Patients removed from positive pressure ventilation were more likely to undergo mobilization on average 11 days earlier than those that remained intubated. Although data suggest that application of early mobilization is safe and feasible, its impact on mortality is not well defined. Limited studies have prospectively assessed the effect of early mobilization on patients receiving ECMO, but suggest a benefit [18,19,20,21].

In conclusion, management of patients with refractory respiratory failure on ECMO without positive pressure ventilation is a promising and increasingly utilized therapeutic strategy. The rationale of endotracheal extubation is based on minimizing exposure to the known complications of positive-pressure ventilation [22] and supporting early mobilization [21]. In this study, we demonstrated that patients with ARDS removed from positive pressure ventilation had increased survival to hospital discharge, increased ventilator free days, and earlier mobilization. While observed differences in our study are likely reflective of patient selection, they should form the basis for future studies to confirm and expand on the potential therapeutic benefits of this management approach.

## Figures and Tables

**Figure 1 jcm-10-00251-f001:**
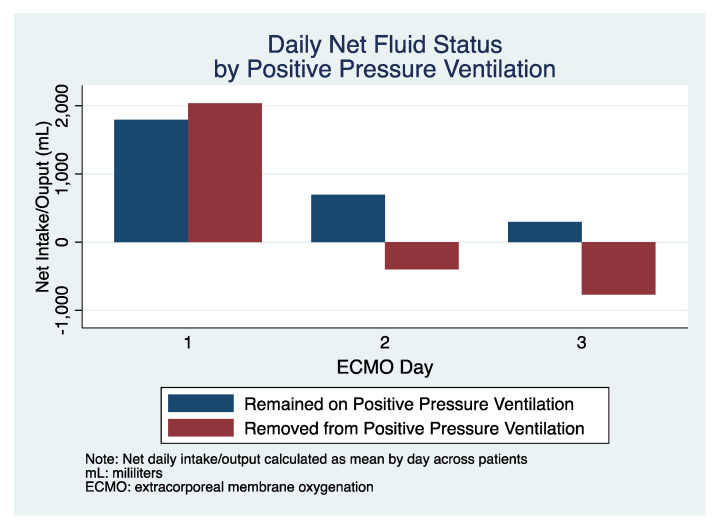
Daily net fluid status over the first three days after ECMO cannulation in patients who remained on positive pressure ventilation (PPV) (blue) and those removed from PPV (red).

**Figure 2 jcm-10-00251-f002:**
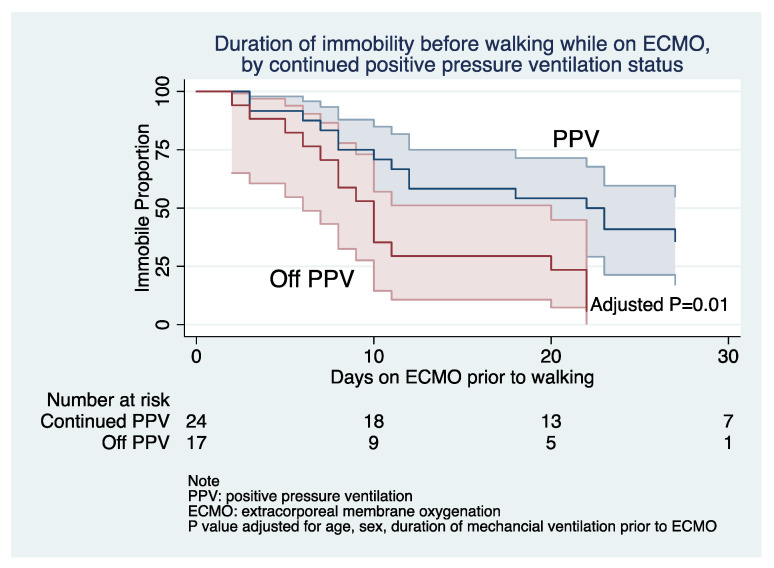
Proportion of patients over time remaining immobilized (not walking) on ECMO between groups; PPV (blue) and patients removed from PPV (red).

**Table 1 jcm-10-00251-t001:** Baseline patient characteristics of patients on extracorporeal membrane oxygenation (ECMO).

	All (*n* = 65)	Presence of Positive Pressure Ventilation on ECMO (*n* = 48)	Absence of Positive Pressure Ventilation on ECMO (*n* = 17)	*p*-Value
Age—yr	44.3 *±* 4.05	44.5 *±* 2.4	43.8 *±* 4.1	*p =* 0.887
Female Sex—no. (%)	19 (29.2)	16 (33.3)	3 (17.7)	*p =* 0.222
Male Sex—no. (%)	46 (70.8)	32 (66.7)	14 (82.4)	*p =* 0.222
Medical Problems—no. (%)				
End-Stage Renal Disease	1 (1.54)	0 (0.0)	1 (5.9)	*p =* 0.090
Cancer (active or observation)	2 (3.1)	1 (2.08)	1 (5.9)	*p =* 0.436
Stroke and/or TIA ^1^	3 (4.62)	2 (4.2)	1 (5.9)	*p =* 0.772
Hypertension	25 (38.5)	18 (37.5)	7 (41.2)	*p =* 0.789
Hyperlipidemia	11 (16.9)	9 (18.8)	2 (11.8)	*p =* 0.509
Diabetes	7 (10.8)	5 (10.4)	2 (11.8)	*p =* 0.878
Coronary artery disease	10 (15.4)	8 (16.7)	2 (11.8)	*p =* 0.630
Congestive heart failure	7 (10.8)	4 (8.3)	3 (17.7)	*p =* 0.287
Respiratory disease (COPD ^2^, Asthma)	19 (29.2)	15 (31.3)	4 (23.5)	*p =* 0.548
DVT ^3^ or PE ^4^	1 (1.5)	1 (2.1)	0 (0.0)	*p =* 0.549
Seizures	0 (0)	0 (0)	0 (0)	n/a
Cirrhosis of the liver	0 (0)	0 (0)	0 (0)	n/a
Unknown	2 (3.1)	2 (4.2)	0 (0)	*p =* 0.393
None	7 (10.77)	5 (10.4)	2 (11.8)	*p =* 0.578

^1^ TIA, transient ischemic attack; ^2^ COPD, chronic obstructive pulmonary disease; ^3^ DVT, deep vein thrombosis; ^4^ PE, pulmonary embolus.

**Table 2 jcm-10-00251-t002:** Extracorporeal membrane oxygenation (ECMO) and ventilator variables.

	All (*n* = 65)	Presence of Positive Pressure Ventilation on ECMO (*n* = 48)	Absence of Positive Pressure Ventilation on ECMO (*n* = 17)	*p*-Value
Murray Score—mean	1.34 *±* 0.3	1.04 *±* 0.3	2.5 *±* 0.6	*p =* 0.031
ECMO Duration—days	14.1 *±* 2.2	13.7 *±* 3.3	14.7 *±* 2.3	*p =* 0.833
Duration of MV ^1^ prior to ECMO—days	5.23 *±* 0.95	5.0 *±* 1.1	6.1 *±* 2.1	*p =* 0.634
Ventilator Days—mean	18.8 *±* 2.6	16.9 *±* 2.9	23.9 *±* 5.8	*p =* 0.240
Ventilator Free Days				
At day 30—mean	7.66 *±* 1.5	5.92 *±* 1.6	12.5 *±* 3.2	*p =* 0.049
At day 60—mean	21.2 *±* 3.3	15.5 *±* 3.7	36.9 *±* 5.3	*p =* 0.003
Ventilator settings (average of first three days on ECMO)—mean				
PaO_2_ ^2^/FiO_2_ ^3^	1.0 *±* 0.6	1.1 *±* 0.7	.87 *±* 0.12	*p =* 0.183
PEEP	11.8 *±* 0.7	11.7 *±* 0.9	12.1 *±* 0.9	*p =* 0.779
Respiratory Rate	20.7 *±* 0.8	20.7 *±* 0.97	20.9 *±* 1.3	*p =* 0.916
Peak Inspiratory Rate	12.7 *±* 0.6	12.7 *±* 0.8	12.7 *±* 0.9	*p =* 0.982
Compliance	24.5 *±* 3.3	21.8 *±* 2.7	35.3 *±* 12.0	*p =* 0.101

^1^ MV, mechanical ventilation; ^2^ PaO_2_, partial pressure of oxygen; ^3^ FiO_2_, fraction of inspired oxygen.

**Table 3 jcm-10-00251-t003:** Primary and secondary measured outcomes.

	All (*n* = 65)	Presence of Positive Pressure Ventilation on ECMO (*n* = 48)	Absence of Positive Pressure Ventilation on ECMO (*n* = 17)	*p*-Value
Time from cannulation to mobilization—days	14.02 *±* 2.3	18.3 *±* 3.4	7.1 *±* 1.5	*p =* 0.015
Time from cannulation to extubation—days	10.5 *±* 2.3	11.5 *±* 3.0	9.2 *±* 3.7	*p =* 0.626
Time from extubation to discharge—days	10.9 *±* 1.2	11.5 *±* 1.7	10.1 *±* 2.0	*p =* 0.599
Total Fluid Balance (first three days)—mL	+ 591 (341)	+761 (410)	+176 (624)	*p =* 0.441
Riker SAS ^1^ (highest first three days)	3.01 *±* 0.2	2.8 *±* 0.2	3.6 *±* 0.3	*p =* 0.049
Average time until GCS 11T or greater—days	9.21 *±* 1.2	8.1 *±* 2.2	9.7 *±* 1.5	*p =* 0.553
LOS ^2^, if survived to discharge—days (mean)	37.0 (4.3)	41.6 (6.4)	31.0 (5.0)	*p =* 0.222
Survived to hospital discharge—no. (%)	39 (60.0)	22 (45.8)	17 (100.0)	*p* < 0.001
Discharge Location—no. (%)				
Home	15 (38.5)	8 (36.4)	7 (41.2)	*p =* 0.674
LTAC ^3^	8 (20.5)	4 (18.2)	4 (23.5)	
SNF ^4^	1 (2.6)	0 (0)	1 (5.9)	
Rehab	13 (33.3)	8 (36.4)	5 (29.4)	
VA	0 (0)	0 (0)	0 (0)	
Other Non-VA Hospital	1 (2.6)	1 (4.6)	0 (0)	
Other	1 (2.6)	1 (4.6)	0 (0)	

^1^ SAS, Sedation Agitation Score; ^2^ LOS, length of stay; ^3^ LTAC, long term acute care; ^4^ SNF, skilled nursing facility.

## Data Availability

Data code is available from the corresponding author upon reasonable request.

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
