# Peer review of "Characteristics of Patients Managed without Positive Pressure Ventilation While on Extracorporeal Membrane Oxygenation for Acute Respiratory Distress Syndrome"

_jcm, 2021, doi:10.3390/jcm10020251_

Round 1

Reviewer 1 Report

I read and reviewed your manuscript interestingly. But I found several fitfalls.

You described that off-PPV showed earlier mobilization (out-of-bed) and better survival discharge. On the other hands, ventilator free days (VFD60) was shorter in the group of off-PPV, and hospital length (LOS) did not differ. Longer ECMO duration, ventilator days prior to ECMO, and higher Murray score and lower PaO2/FiO2 in the group of off-PPV were meant more severe conditions. I couldn't understand this discrepancy. Because survival benefits can usually come from other minor outcome enhancements. You must explain reason for discrapancy between survival outcomes and variables, and others.

The manuscript is titled ECMO for ARDS, but in the data, number of VA ECMO is more than VV ECMO. Usually VV ECMO is more frequently used in adjunction for ARDS. If there were numbers of VA ECMO for cardiogenic causes, it might be explained shorter ECMO duration and higher PF ratio in the on-PPV group. Thus, reasons for ECMO must be revealed manuscript.

Your large cohort for awake ECMO proved that off-PPV is effective to earlier mobilization, which leads to better survival. Therefore, I recommend obvious causal classification for ECMO, and an analysis based on the reasons for the ECMO, or VV ECMO alone.

Author Response

Response to Reviewers:

We would like thank the editor for the opportunity to have our paper reviewed. We had the opportunity to review the reviewers’ comments on our article. The comments are much appreciated. We think that they have significantly improved the quality of the manuscript.

We have addressed each comment below, and cited the changes we made to the text for ease of reading.

Reviewer: 1

Comments to Author

You described that off-PPV showed earlier mobilization (out-of-bed) and better survival discharge. On the other hand, ventilator free days (VFD60) was shorter in the group of off-PPV, and hospital length (LOS) did not differ. Longer ECMO duration, ventilator days prior to ECMO, and higher Murray score and lower PaO2/FiO2 in the group of off-PPV were meant more severe conditions. I couldn't understand this discrepancy. Because survival benefits can usually come from other minor outcome enhancements. You must explain reason for discrepancy between survival outcomes and variables, and others.

The manuscript is titled ECMO for ARDS, but in the data, number of VA ECMO is more than VV ECMO. Usually VV ECMO is more frequently used in adjunction for ARDS. If there were numbers of VA ECMO for cardiogenic causes, it might be explained shorter ECMO duration and higher PF ratio in the on-PPV group. Thus, reasons for ECMO must be revealed manuscript.

Your large cohort for awake ECMO proved that off-PPV is effective to earlier mobilization, which leads to better survival. Therefore, I recommend obvious causal classification for ECMO, and an analysis based on the reasons for the ECMO, or VV ECMO alone.

RESPONSE: We sincerely appreciate the reviewer comments.

  • To make the groups more homogenous, we have limited to the analysis to patients who were diagnosed with ARDS and treated on VV ECMO. We have redone our analysis and updated our sample appropriately, and have presented the results.
  • We sincerely appreciate the astute observation of the reviewer on the discrepancy in VFDs and hospital LOS between the two groups. By limiting to patients only on VV ECMO for ARDS, and re-running the analysis, we have updated our results to demonstrate the patients who were extubated on ECMO had significantly higher numbers of VFDs at 30 and 60 days.

Reviewer 2 Report

1. Stylistic consideration: I would consider streamlining the abstract. It is a bit wordy.

2. Inconsistent hyphenations: sometimes non-significant is hyphenated and sometimes it is not, non PPV should be hyphenated (see lines 144, 155)

3. Under the introduction: the last paragraph lines 71-74 is talking about results and should be removed from the introduction. In that last paragraph of the introduction, I would talk about your hypothesis, aims, etc.

4. The authors state that only patients in the CVICU were included in the study, please clarify if all patients treated with ECMO are only treated in the CVICU or not (many centers differ as to CVICU, MICU, etc).

5. Some of the abbreviations in the tables are not defined. Many of these are very common and many providers would what there are. Examples: table 1 --> TIA, COPD, DVT, PE.    Table 2--> Pao2, FiO2.

6.  Under the discussion: Lines 176-179 is rehashing results without adding new information or connecting it to other studies. IE... let the readers now how those results are similar or different from other published studies (if there are any)

Author Response

Response to Reviewers:

We would like thank the editor for the opportunity to have our paper reviewed. We had the opportunity to review the reviewers’ comments on our article. The comments are much appreciated. We think that they have significantly improved the quality of the manuscript.

We have addressed each comment below, and cited the changes we made to the text for ease of reading.

Reviewer: 2

Comments to Author

Stylistic consideration: I would consider streamlining the abstract. It is a bit wordy.

RESPONSE: We thank the reviewer for highlighting this opportunity to make our abstract more concise. After further review we removed certain elements, such as the VA ECMO component and a few reported results like demographic data.

Inconsistent hyphenations: sometimes non-significant is hyphenated and sometimes it is not, non PPV should be hyphenated (see lines 144, 155)

RESPONSE: Thank you for pointing out this grammatical error. In both instances this has been corrected.

Under the introduction: the last paragraph lines 71-74 is talking about results and should be removed from the introduction. In that last paragraph of the introduction, I would talk about your hypothesis, aims, etc.

RESPONSE: We agree and as such removed that paragraph and concluded the introduction with:

While there is literature to suggest the safety and feasibility of extubation while on venovenous (VV) ECMO, studies are limited to ≤12 patients [6,11,12], and or are among patients with chronic respiratory failure as a bridge to transplant [13,14]. In this study we look to describe the characteristics of a large group of patients with ARDS managed on ECMO who achieved breathing without the support of positive pressure ventilation.

The authors state that only patients in the CVICU were included in the study, please clarify if all patients treated with ECMO are only treated in the CVICU or not (many centers differ as to CVICU, MICU, etc).

RESPONSE: We agree this clarification is important given the variable nature of different practice settings at other institutions. We made the following edits:

The University of Utah Hospital is a tertiary referral medical center for Utah, Wyoming, Idaho and Nevada. The medical center’s Cardiovascular Intensive Care Unit (CVICU) serves an integral part of the region’s peri-cardiac and mechanical circulatory support intervention site. The CVICU serves as the specific unit responsible in the care of all ECMO cases at this institution. Patients who were admitted to this service and placed on ECMO during their admission were the subjects of interest.

Some of the abbreviations in the tables are not defined. Many of these are very common and many providers would what there are. Examples: table 1 --> TIA, COPD, DVT, PE.    Table 2--> Pao2, FiO2.

RESPONSE: We thank you for highlighting this oversight and have made note of the abbreviations in the footnotes of the tables.

  1. Under the discussion: Lines 176-179 is rehashing results without adding new information or connecting it to other studies. IE... let the readers know how those results are similar or different from other published studies (if there are any)

RESPONSE: We thank the reviewer for this insight. We have attempted to highlight the benefit of early mobilization based on current literature but recognize that there are no prospective studies yet to demonstrate any mortality benefit.

Round 2

Reviewer 1 Report

I read your revised manuscript well.

It was large cohort for awake ECMO, and outcomes were documented well in the results.

You mentioned that benefit for ECMO without PPV was based on ventilator-associated complication in the conclusion.
However, your result did not showed any safety outcomes, such as ventilator-associated pneumonia or pneumothorax.
Therefore, the conclusion will must be modified.

Author Response

RESPONSE TO REVIEWERS

REVIEWER:

I read your revised manuscript well.

It was large cohort for awake ECMO, and outcomes were documented well in the results.

You mentioned that benefit for ECMO without PPV was based on ventilator-associated complication in the conclusion.
However, your result did not showed any safety outcomes, such as ventilator-associated pneumonia or pneumothorax.
Therefore, the conclusion will must be modified.

RESPONSE:

We thank the reviewers for their comment. We have modified the conclusion accordingly:

“In conclusion, management of patients with refractory respiratory failure on ECMO without positive pressure ventilation is a promising and increasingly utilized therapeutic strategy. The rationale of endotracheal extubation is based on minimizing exposure to the known complications of positive-pressure ventilation [22] and supporting early mobilization [23]. In this study, we demonstrated that patients with ARDS removed from positive pressure ventilation had increased ventilator free days, and earlier mobilization. While observed differences in our study are likely reflective of patient selection, they should form the basis for future studies to confirm and expand on the potential therapeutic benefits of this management approach.”